# Physiologically Based Biopharmaceutics Modeling of Food Effect for Basmisanil: A Retrospective Case Study of the Utility for Formulation Bridging

**DOI:** 10.3390/pharmaceutics15010191

**Published:** 2023-01-05

**Authors:** Tejashree Belubbi, Davide Bassani, Cordula Stillhart, Neil Parrott

**Affiliations:** 1Pharmaceutical Research & Early Development, Roche Innovation Center Basel, F. Hoffmann—La Roche Ltd., 4070 Basel, Switzerland; 2Pharmaceutical Research & Development, Formulation & Process Sciences, F. Hoffmann—La Roche Ltd., 4070 Basel, Switzerland

**Keywords:** food effect, oral absorption, PBBM, immediate-release formulation, poorly water-soluble compound, GastroPlus™

## Abstract

Basmisanil, is a lipophilic drug substance, exhibiting poor solubility and good permeability (BCS class 2). A validated physiologically based biopharmaceutics model (PBBM) has been previously described for tablets dosed in the fed state. The PBBM captured the less than proportional increases in exposure at higher doses well and indicated that absorption was dissolution rate-limited below 200 mg while solubility was limiting for higher doses. In this study, a model for dosing in the fasted state is described and is verified for simulation of the food effect where exposures were ~1.5 fold higher when a 660 mg tablet was given with food. The model is then applied to simulate the food effect for a granules formulation given at a lower dose (120 mg). The food effect at the lower dose was reasonably simulated with a ratio of simulated/observed food effect of 1.35 for Cmax and 0.83 for AUC. Sensitivity analysis was carried out for uncertain model parameters to confirm that the model could predict the magnitude of the positive food effect with moderate to high confidence. This study suggests that a verified PBBM can provide a useful alternative to a repeat food effect study when formulation changes are minor. However, there is need for further evaluation of the approach and a definition of what formulation changes are minor in this context. In addition, this work highlights some uncertainties in the handling of solubility in PBBM, in particular around temperature dependency of solubility and the parameterization of bile salt solubilization using measurements in biorelevant media.

## 1. Introduction

Since the first examples appeared approximately 15 years ago [1], publications describing prediction of the impact of food on the absorption of orally administered drugs using generic physiologically based pharmacokinetic (PBPK) modeling platforms have become increasingly frequent. Industry consortia have invested in model validation efforts [2,3,4,5] but due to the many potential complexities in food effect mechanisms [6], the confidence is still low and the impact for regulatory questions remains limited. In a recent review [7], Emami-Riedmaier proposed a *middle-out* PBPK modelling approach to deliver higher confidence food effect projections following formulation or dose changes. In such an approach, both the fasted and fed state models would be verified using a clinical anchor study to allow extrapolation to additional scenarios and waiver of uninformative clinical studies.

In this study we have applied this approach retrospectively for basmisanil, a lipophilic biopharmaceutics classification system (BCS) class 2 substance exhibiting poor solubility and good permeability. In particular for basmisanil a mechanistic understanding of the effect of food on oral absorption is valuable to guide further formulation development for dosing in specific patient populations such as children. A physiologically based biopharmaceutics model (PBBM) [8] for basmisanil has been described previously by Stillhart et al. [9] who validated a PBBM incorporating biorelevant in vitro dissolution data using single-dose pharmacokinetic data from multiple studies and developed an in vitro-in vivo correlation (IVIVC) using mechanistic deconvolution for tablet and granules formulations of micronized drug substance. This previously described PBBM was built and verified only for dosing in the fed state which was identified as showing a ~2-fold increased exposure compared to the fasted state. In the current paper, we extend the PBBM to cover basmisanil when dosed in the fasted state and apply it for the simulation of the food effect for tablets used in phase 1 studies and then for a granules formulation developed for the market. Thus, this study evaluates the proposed *middle-out* strategy for PBBM verification [7] and explores the potential to project food effect across different formulations of a drug substance.

## 2. Materials and Methods

### 2.1. API and Formulation Properties

The physicochemical properties of the drug substance are listed in Table 1. The effective human jejunal permeability of 3.75 × 10^−4^ cm/s was obtained from permeability measurement across Caco-2 cells. The previously described method [10] was used to convert the in vitro permeability to an estimate of human jejunal permeability. Drug solubility was determined experimentally at room temperature in aqueous buffer at a range of pH and also in biorelevant media at 37 °C. Three biorelevant media namely, simulated gastric fluid (SGF) pH 1.6, fasted-state simulated intestinal fluid (FaSSIF) pH 6.5, and fed-state simulated intestinal fluid (FeSSIF) pH 5.0 were used for the measurements of biorelevant solubilities [11].

Basmisanil was formulated in micronized form and exhibited a D90 of less than 10 µm. Uncoated immediate-release tablets were used for phase 1 clinical studies and subsequently a granules-in-sachet formulation was developed for phase 2 studies. The details of the formulation preparation and the excipients used have been outlined previously [9].

### 2.2. Clinical Pharmacokinetic Data of Basmisanil

Preclinical PBPK modeling had projected a significant positive food effect potential for basmisanil. Therefore, the first in human single ascending dose study was performed in the fed state. However, at the highest anticipated effective dose of 660 mg a food effect arm was performed in 8 subjects. Subjects were dosed under fed or fasted conditions (an overnight fast) in a random order. The mean age was 74 years, 8 males and mean BMI was 23.5. For the fed state, the dose was given with 250 mL of still water within 5 min after completion of a standardized breakfast containing an average of 594 kcal, of which 27% was fat, i.e., a low-fat meal. Plasma concentrations were measured pre-dose, as well as 0.5, 1, 1.5, 2, 2.5, 3, 4, 6, 8, 11, 14, 24, 36, 48, and 72 h post-dose.

In a separate study, the systemic pharmacokinetics of basmisanil was determined in six healthy adult volunteers. The mean age was 41 years, 6 males, 4 of white or Asian race, mean body weight was 83 kg and mean BMI was 26.5. The subjects were administered an intravenous microdose of 0.1 mg [^13^C]-labeled API as an aqueous solution by a constant rate infusion over 15 min starting 3.75 h after the administration of an oral capsule dose of 160 mg. Plasma concentrations were measured pre-dose, as well as 5, 15, 20, 25, 30, 45 min, and 2.25, 4.25, and 8.25 h post-dose. 

For phase 2 studies a granules-in-sachet formulation was developed, and the interest had switched to a 120 mg bid dosing regimen. The granules formulation was evaluated in a human PK study performed in 18 healthy adult volunteers in both fasted and fed conditions. The mean age of the volunteers was 36 years, 10 males and 8 females all of white race. Mean body weight was 78 kg and mean BMI was 25.8. For the fed state dosing, single doses were administered within 30 min of the start of a standardized breakfast containing an average of 895 kcal of which 61% was fat, i.e., a high-fat meal. Granules were sprinkled onto a tablespoon of apple sauce and swallowed followed by 240 mL of still water. Plasma concentrations were determined pre-dose, as well as 0.5, 1, 1.5, 2, 3, 4, 5, 6, 8, 10, 12, 16, 24, 36, 48, and 60 h post-dose. All clinical studies were reviewed and approved by independent ethics committees and conducted in full conformance with the International Conference on Harmonisation (ICH) E6 guideline for Good Clinical Practice (GCP) and the principles of the Declaration of Helsinki.

### 2.3. Construction of the Baseline GastroPlus™ Model

A two-compartment PK model was fitted to the mean plasma concentration vs. time profile to describe the disposition kinetics of basmisanil. The plasma concentration vs. time profile was obtained after a single intravenous administration to 6 healthy male volunteers. The physicochemical model input values and the derived pharmacokinetic parameters were integrated together in the GastroPlus™ (Simulations Plus, Lancaster, CA, USA) model (Table 1). It was assumed that intestinal and hepatic first pass extraction were negligible, due to the very low extraction ratio of <5% of the liver blood flow.

For the GastroPlus™ model, in vivo drug release was predicted using the Johnson dissolution model [10]. “IR tablet” for uncoated tablets and “IR suspension” for granules in sachet formulation were chosen as the dosage form assuming immediate-release kinetics all formulations. The drug particle size distribution was included (Table 1) assuming that particles were distributed into eight bins with a constant radius in each bin.

### 2.4. Food Effect Modeling Approach

The intestinal solubility model in GastroPlus™ takes into account the data on the changing concentrations for bile salts along the gastrointestinal tract [11,12] as well as the solubility enhancement due to bile salt solubilization in the fed and fasted states [13] to derive full biorelevant regional solubility profiles. The relationship between the concentration of bile salt in the medium and the enhancement of solubility is a compound-specific parameter characterized as a solubilization ratio (SR) [13]. Two SR values were estimated within GastroPlus™ based on (i) FaSSIF solubility (bile salt concentration 3 mM) and (ii) FeSSIF (bile salt concentration 15 mM) as reported in Table 1 and these two values were used in the GastroPlus™ fasted and fed state models, respectively to model the food effect.

In this work we applied the fasted and fed state physiological ACAT models which are included in GastroPlus™ version 9.8.2. These models account for food related increases of gastric emptying time, volume and pH, increased hepatic blood flow, decreased upper intestinal pH, and increased bile salt concentration. For the fed state, gastric emptying follows a zero-order kinetic where the gastric emptying time is calculated based on a linear correlation to the caloric content of the meal. In addition, the bile salt concentrations in the intestinal compartments are calculated based on the percentage of fat in the meal using a correlation function developed by Simulations Plus based on published data. In this work, we adjusted the fed state ACAT models for each study using the reported meal calorie and fat content.

The strategy applied for the food effect model verification and optimization with clinical data was as proposed by Riedmaier et al. [14]. Following the decision tree (see Appendix A) proposed in that work we first checked the fasted state simulations to ensure agreement to clinical data within 0.8- to 1.25-fold of observed before predicting the fed state exposures.

### 2.5. Parameter Sensitivity Analysis

A parameter sensitivity analysis (PSA) was conducted for both the fasted and the fed states of the 660 mg uncoated tablets and 120 mg granules. The sensitivity to rate and extent of oral absorption expressed as Cmax and AUC (0-inf) was measured. The parameters tested were permeability, particle size, bile salt solubilization ratio and stomach pH. The parameters were varied with logarithmic spacing (10 steps).

## 3. Results

### 3.1. Simulations for Uncoated Tablet Formulation at a Dose of 660 mg

Firstly, the model for dosing in the fasted state was constructed and verified with clinical pharmacokinetic data for a dose of 660 mg given as uncoated tablets. A solubility vs. pH profile was constructed using the solubility of 8 μg/mL measured in simulated gastric fluid (SGF) at a pH of 1.6, together with the measured aqueous buffer solubility of 0.001 μg/mL at values of pH 3 and higher. When using the measured basic pKa of 2.07 the aqueous solubility vs. pH profile generated in GastroPlus™ described the measured data well as shown in Figure 1. The only fitted parameter was the solubility factor of 8.

The next step was to incorporate measurements of biorelevant solubility in simulated fasted state intestinal media. As the aqueous solubility was measured at 25 °C, the default GastroPlus™ temperature correction was first applied to estimate the solubility at 37 °C. The corrected intrinsic solubility was found to be 9.5 µg/mL. Then, the solubilization effect of bile salts was captured by using the measured FaSSIF solubility of 10 µg/mL at pH 6.5. A bile salt solubilization ratio of 966 was estimated. After applying these model adjustments, the simulated profile for a 660 mg dose in the fasted state was in good agreement with the mean observed profile (Figure 2 and Table 2).

According to the decision tree [14], since the simulated AUC and Cmax parameters for the fasted state were within 0.8 to 1.25 of observed values we then proceeded to simulate the fed state. There were no further model optimizations, the only model changes were to use the measured FeSSIF solubility of 32 µg/mL at pH 5 to estimate a bile salt solubilization ratio of 8754 (9 times higher than the value for the fasted state simulation). This was used as an input to the GastroPlus™ fed state model which accounted for the calories and fat content of the actual used meal (594 kcal, 27% fat) to estimate the appropriate gastric emptying time and bile salt concentrations. The simulated profile captured the positive food effect. However, simulations underestimated the observed mean values for Cmax and AUC by ~25%. The simulated/observed ratio for Cmax and AUC were 0.764 and 0.786 in the fed state. According to the decision tree this indicates a moderate level of confidence in this model for food effect projection since the fed Cmax and AUC were within 2-fold of observed but were not within 0.8 to 1.25 (high confidence).

### 3.2. Model Parameter Sensitivity Analysis at a Dose of 660 mg

Parameter sensitivity analyses was used to explore which of the uncertain model inputs most strongly influenced the fasted and fed state simulations (Figure 3, Table 3). For the fasted state, the baseline parameters led to a simulated profile which matched the observed profile well and resulted in a simulated fraction absorbed of ~30%. The simulated pharmacokinetics were not highly sensitive to any of the explored inputs. Thus, changes in bile salt solubilization ratio, particle size and stomach pH (+/− 3-fold, +/− 3-fold, 0.5 to 5, respectively) only affected AUC and Cmax by less than 20%. Changes in permeability (+/− 2-fold) were more sensitive, affecting the Cmax and AUC by up to 50%. In the fed state a similar sensitivity as in fasted state was seen for permeability. Sensitivity to stomach pH was negligible while particle size sensitivity was similar to that seen in the fasted state. However, the sensitivity to fed state bile salt solubilization ratio was much higher than in the fasted state with an increase of 73% in simulated Cmax for a 3-fold change in solubilization ratio.

### 3.3. GastroPlus™ Model Simulations for Granules at a Dose of 120 mg

The model verified with fasted and fed state simulations at a dose of 660 mg as uncoated tablets was applied to project the food effect for a dose of 120 mg given as granules. There were no changes made for the fasted state model while the only change to the model for the fed state was to set the meal to a calorie content of 895 kcal of which 61% was fat. Simulated profiles were close to the observed data for both fasted and fed states (Figure 4 and Table 4). The simulated/observed ratio for Cmax and AUC were 1.2 and 1.1, respectively, in the fed state and 0.96 and 1.3, respectively, in the fasted state.

The simulated and observed food effect ratios (fed/fasted) are summarized in Table 5. For the uncoated tablets at a dose of 660 mg the positive food effect of ~2-fold was slightly underestimated for both Cmax and AUC. At the lower dose of 120 mg as granules the positive food effect was reduced to ~1.5-fold and this was reasonably captured.

Parameter sensitivity analyses were performed at a dose of 120 mg and showed an overall reduced sensitivity of model simulations compared to the 660 mg dose when using the same range of explored input parameters (see Appendix A).

## 4. Discussion

According to a recent commentary [7], when the driving mechanism of the food effect for a specific drug is related to changes in its solubility in the gastrointestinal fluids the quantitative food effect is predictable with high (within 1.25-fold) to moderate (within 2-fold) confidence via PBBM. In this study with basmisanil, a PBBM was constructed in a bottom-up manner, integrating measured in vitro data. Biorelevant solubility measurements in SGF, FaSSIF and FeSSIF together with effective human jejunal permeability scaled from measurements across Caco-2 cells were integrated into the model according to best practices as defined in a recent cross-industry study [2]. The dissolution rate of basmisanil was modelled based on the measured particle size distribution input into the standard Noyes-Whitney model. Simulation of the fed and fasted state pharmacokinetics was verified with clinical data, and at a dose of 660 mg the ~2-fold positive food effect was predicted with a ratio (predicted/observed) of 0.76 for Cmax and 0.93 for AUC thus falling in the high to moderate confidence range. This is as expected since the mechanism for the food effect was clearly related to the ~3-fold higher solubility in the fed state fluids compared to fasted state fluids (0.032 mg/mL and 0.01 mg/mL, respectively). Considering that the PBBM could capture the food effect at 660 mg moderately well (ratio of predicted to observed food effects of 0.76 and 0.93 for Cmax and AUC, respectively) the model was applied to predict the food effect at a lower dose of 120 mg. In addition to a lower dose this prediction was made for a different formulation, namely granules in a sachet and not for the uncoated tablets used for the initial food effect study. The tablet and granules formulations were similar with respect to qualitative composition and were produced using the same manufacturing technology and the same micronized API. The ratio of (predicted/observed) food effect for the 120 mg dose was 1.35 for Cmax and 0.83 for AUC, again falling in the high to moderate confidence range for predictions.

Development of basmisanil was not changed in any way based on the results of the repeated food effect study and the assessment of food effect for the granules formulation was not meaningfully different to that for the earlier formulation. Thus, in this case PBBM could have provided an adequate alternative to the 2nd food effect study.

Basmisanil is a BCS class 2 molecule, and its positive food effect is typical of lipophilic poorly soluble drugs where the luminal solubility is enhanced by bile salts. Multiple examples have been reported of successful PBBM prediction of food effect for such molecules when incorporating measured solubility in biorelevant media for the fasted and fed states [1,11,15,16].

For basmisanil, the PBBM had been developed pre-clinically and due to the predicted positive food effect, the first-in-human single ascending dose study was performed with food. A food effect arm was performed in that study at a dose estimated from Pharmacokinetic/Pharmacodynamic (PK/PD) considerations to be the highest likely clinical dose for phase 2 studies. That food effect study confirmed the reduced exposures in fasted state and together with the good agreement of simulated and observed PK across the full dose range tested, increased confidence in the PBBM. Since no major formulation changes were made for the granules in sachet formulation it was expected that the PBBM could predict the food effect for the phase 2 formulation.

One uncertainty introduced into the current model was the use of solubility measured at room temperature. For optimal biorelevance, aqueous solubility should be measured at 37 °C however this requires additional experimental effort and in the case of basmisanil our measurements of aqueous solubility vs. pH were made at an early stage at room temperature (25 °C) and measurements were not repeated at 37 °C. This introduced uncertainty in the construction of the PBBM model in the fasted state since it is known that solubility of pharmaceuticals can vary between room temperature and body temperature [17]. GastroPlus™ includes an option to estimate API solubility at 37 °C based on a measurement at a reference temperature and the melting point of the API. For basmisanil, this correction changed the estimated reference aqueous solubility at pH 7 from 1 µg/mL to 9.5 µg/mL and this in turn affected the estimated biorelevant solubilization ratio for basmisanil (see discussion below). We were surprised at this >9-fold predicted change in solubility for basmisanil for a change in temperature of 12 °C and so we investigated predictions of this model for 13 drug-like compounds with reported temperature-dependent solubility. GastroPlus™ was able to capture the experimental data very well for two compounds. For the others, GastroPlus™ slightly overestimated the temperature dependent solubility when compared to the experimental data. (See Appendix A). This is true for molecules of different weight, volume, lipophilicity, and ionization state at pH 7 (apart from the molecular weight, all these physicochemical descriptors are not considered by the formula implemented in GastroPlus). Since the measurement temperature for aqueous solubility can significantly affect PBBM construction it is recommended that this should be systematically reported in future publications.

Although integrating biorelevant solubility measurements into PBBM can capture the food effect well for a BCS 2 molecule like basmisanil, there is still room for improvement in handling of the effects of food on luminal fluids and solubility. The presence of bile salts in the intestinal fluids often enhances the solubility of lipophilic molecules as compared to aqueous media [18]. PBBM can account for this enhancement of solubility in the fed and fasted states due to bile salt solubilization and also consider the changing physiological concentrations of bile salts in different regions of the gastrointestinal tract to derive regional solubility profiles [11,12]. The GastroPlus™ platform uses the compound-specific parameter called solubilization ratio [13,18], whereas SimCYP™ uses bile micelle: water partition coefficients (Km:w) for the ionized and un-ionized drug [17]. While it is well acknowledged that solubility measured in biorelevant media such as FeSSIF [18] and FaSSIF should be used to build a PBBM, the exact method used to incorporate these data can vary and is often not reported in PBBM publications. Establishment of best practices for this important step could be helpful to increase confidence in predictions. For basmisanil we explored different methods to fit the ***SR*** parameter within the GastroPlus™ platform. This fitting is made by applying the equation below to the entered aqueous solubility, FaSSIF and FeSSIF values:$$C_{s,bile}=C_{s,aq}+Sc_{aq}\times SR\times M_{w}\times {\left[bile\right]}^{N}$$
where ***C_s,aq_*** is the aqueous solubility at given pH, ***C_s,bile_*** is the solubility in the presence of bile salts at concentration [***bile***] and the same pH as ***C_s,aq_***, ***Sc_aq_*** is the aqueous solubilization capacity calculated as the ratio of moles of drug to moles of water at a concentration equal to aqueous solubility, ***M_w_*** is the drug molecular weight, ***SR*** is the bile salt solubilization ratio and ***N*** is the ***SR*** exponent. For basmisanil, a good fit of values of ***SR*** and ***N*** which reconciled the measured values of ***C_s,bile_***, ***C_s,aq_*** and [***bile***] for both fed and fasted states was not possible and so we used individually fitted ***SR*** values for fasted and fed states.

## 5. Conclusions

This study suggests that PBBM can provide a useful alternative to a repeat food effect study when formulation changes are minor. However, for increased confidence in the constructed PBBM the aqueous solubility should be measured at 37 °C since prediction of the temperature dependency can introduce error. Additionally, further work on the optimal parameterization of bile salt solubilization using measurements in biorelevant media is recommended.

## Figures and Tables

**Figure 1 pharmaceutics-15-00191-f001:**
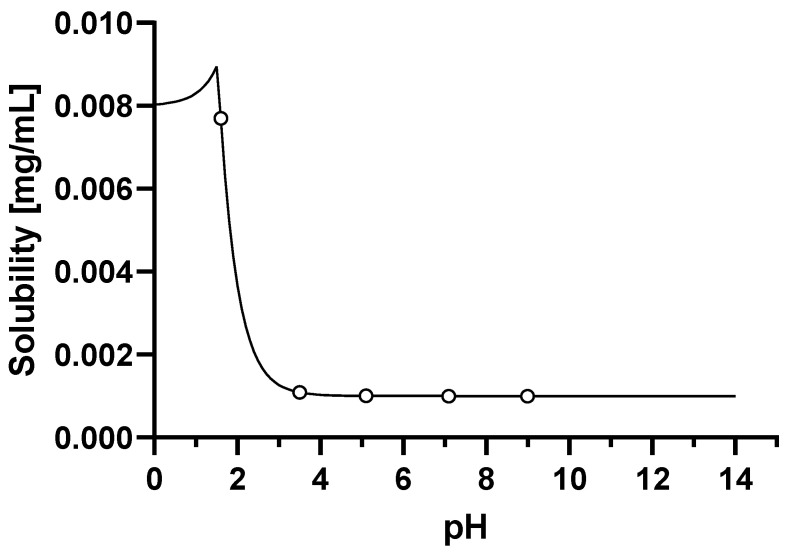
Solubility vs. pH profile of basmisanil (fitted parameters are represented by a continuous line, experimental values are represented by circles).

**Figure 2 pharmaceutics-15-00191-f002:**
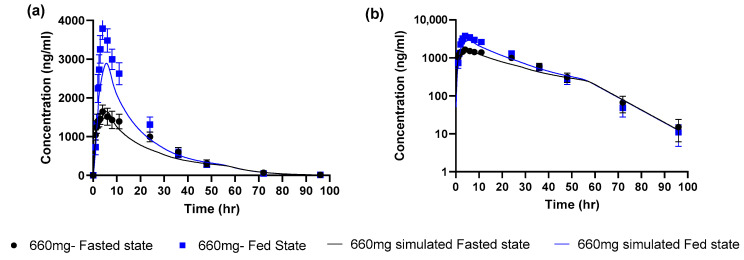
Observed (mean ± SEM) and simulated profiles for 660 mg uncoated tablet (**a**) linear scale (**b**) log scale.

**Figure 3 pharmaceutics-15-00191-f003:**
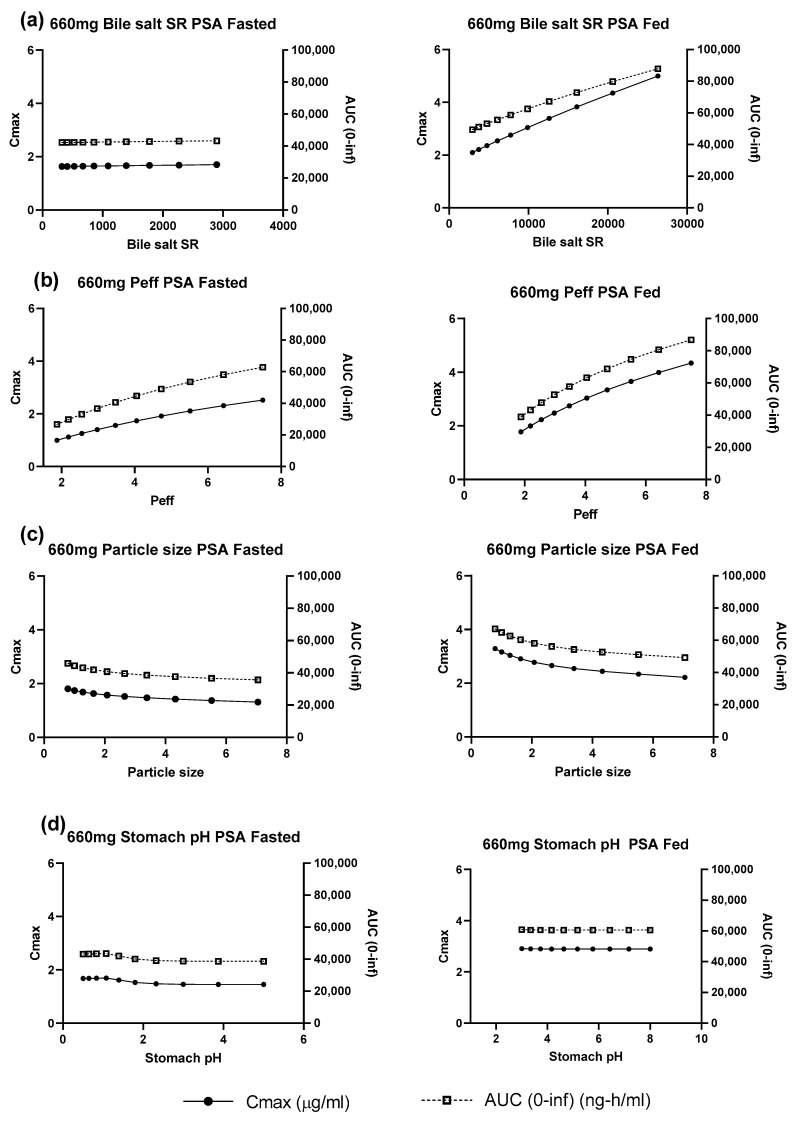
Parameter sensitivity analysis for 660 mg uncoated tablets under simulated fasted and fed states (**a**) bile salt solubilization ratio (**b**) permeability (**c**) particle size (**d**) stomach pH.

**Figure 4 pharmaceutics-15-00191-f004:**
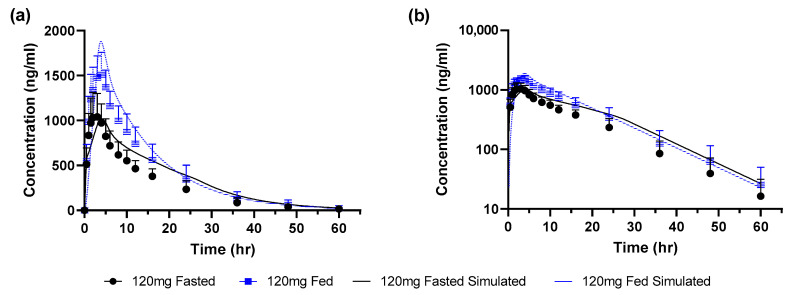
Observed (mean ± SEM) and simulated profiles for 120 mg granules (**a**) linear scale (**b**) log scale.

**Table 1 pharmaceutics-15-00191-t001:** Physicochemical characteristics and kinetics of disposition of Basmisanil.

Parameter *	Value
Molecular weight	445.5 g/mol
pK_a_	2.07 (base)
logD (pH 7.4)	1.86
Melting point	148.3 °C
Blood/plasma concentration ratio Fraction of drug unbound in plasma	0.59 5.6%
Solubility at 25 °C	
Aqueous buffer pH 1–9 Solubility at 37 °C	0.001 mg/mL
SGF pH 1.6	0.008 mg/mL
FaSSIF pH 6.5	0.010 mg/mL
FeSSIF pH 5	0.032 mg/mL
Particle size distribution	
D10	1.4 μm
D50	4.7 μm
D90	10.1 μm
Effective human jejunal permeability	3.75 × 10^−4^ cm/s
Disposition model parameters	
k_12_	1.294 1/h
k_21_	0.979 1/h
k_10_	0.245 1/h
V_c_/kg	0.235 L/kg
V_2_/kg	0.311 L/kg
CL/kg	0.058 L/h/kg
CL_2_/kg	0.304 L/h/kg
Elimination half-life	7.0 h

* pKa: Acid dissociation constant; logD: Distribution coefficient; SGF: Simulated gastric fluid; FaSSIF: Fasted state simulated intestinal fluid; FeSSIF: Fed state simulated intestinal fluid; k_12_ and k_21_: Transfer rate constant from the central to the peripheral compartment and from the peripheral to the central compartment, respectively; k_10_: Elimination rate constant; V_c_ and V_2_: Volumes of distribution of the central compartment and in the elimination phase, respectively; CL and CL_2_: Clearances of the central compartment and in the elimination phase, respectively.

**Table 2 pharmaceutics-15-00191-t002:** 660 mg basmisanil as an uncoated tablet formulation in the fasted and fed state using Johnson dissolution model.

	Fasted State			Fed State		
Parameters	Observed	Simulated	Simulated/Observed	Observed	Simulated	Simulated/Observed
Cmax (ng/mL)	1649	1648	0.999	3787	2892	0.764
AUCinf (ng.h/mL)	51,400	42,600	0.829	76,900	60,500	0.786
Tmax (h)	4	5.74	1.435	4	5.64	1.410

**Table 3 pharmaceutics-15-00191-t003:** Parameter sensitivity analysis for 660 mg uncoated tablets.

Parameters	Fasted State			Fed State		
	Parameter Range Explored (Baseline)	Cmax (ng/mL)	AUC(0-inf) (ng.h.mL)	Parameter Range Explored (Baseline)	Cmax (ng/mL)	AUC(0-inf) (ng.h.mL)
Peff	1.87–7.5 (**3.75**)	60.4–153	62.6–148	1.87–7.5 (**3.75**)	61.5–150	64.3–143
Particle size (D50)	0.78–7.05 (**2.35**)	109–79.5	108–83.5	0.78–7.05 (**2.35**)	113.7–76.7	111–81.3
Bile salt Solubiliza-tion Ratio	323–2897.1 (**965.7**)	98.9–103	99.5–102	2917–26,300 (**8753.6**)	72.4–173	81.6–145
Stomach pH	0.5–5 (**1.3**)	102–88.1	101–90.8	4–6 (**4.9**)	100–99.9	99.9–99.9

Cmax and AUC are expressed as the percentage of the value for the baseline simulation. Green shading = within 80–125%, red shading = beyond 80–125%.

**Table 4 pharmaceutics-15-00191-t004:** 120 mg basmisanil as granules in the fasted and fed state using Johnson dissolution model.

	Fasted State			Fed State		
Parameters	Observed	Simulated	Simulated/Observed	Observed	Simulated	Simulated/Observed
Cmax (ng/mL)	1043	1006	0.96	1513	1877	1.2
AUCinf (ng.h/mL)	16,010	20,760	1.3	24,050	26,090	1.1
Tmax (h)	3	4.26	1.4	4	3.94	0.99

**Table 5 pharmaceutics-15-00191-t005:** Fed/Fasted ratios for simulated and observed profiles.

Formulation	Simulated		Observed *		Ratio of Simulated/Observed Food Effect	
	Cmax	AUC	Cmax	AUC	Cmax	AUC
660 mg uncoated tablets	1.75	1.42	2.31	1.52	0.76	0.93
120 mg granules	1.86	1.25	1.38	1.5	1.35	0.83

* Geometric means.

## Data Availability

Not applicable.

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
