# Peer review of "Physiologically Based Biopharmaceutics Modeling of Food Effect for Basmisanil: A Retrospective Case Study of the Utility for Formulation Bridging"

_pharmaceutics, 2023, doi:10.3390/pharmaceutics15010191_

Round 1
Reviewer 1 Report
Regarding the manuscript (pharmaceutics-2101046) entitled:
“Physiologically Based Biopharmaceutics Modeling of Food Effect for Basmisanil: A Retrospective Case Study of the Utility for Formulation Bridging”
Comments to the Author
General comment
The manuscript described a model for dosing in the fasted state for simulation of the food effect. The manuscript, in general, is well written but some points should be considered:
1. Please provide the ethics committee approval number and details about the guideline used for the clinical study.
2. Please provide some details about the healthy adult volunteers.
1. What is the main question addressed by the research?
Can PBBM provide a useful alternative to a repeat food effect study when formulation changes are minor?
2. Do you consider the topic original or relevant in the field? Does it
address a specific gap in the field?
Yes, using Baseline GastroPlus™ Model could provide a good tool in the future for the prediction of drug profiles.
3. What does it add to the subject area compared with other published
material?
Few studies reported GastroPlus™ Model as a prediction model but still need more studies using different drugs.
4. What specific improvements should the authors consider regarding the
methodology? What further controls should be considered?
This study was built on a previous study published by Stillhart et al 2017. It is worth to conducted another study using another API with similar properties to validate the model.
5. Are the conclusions consistent with the evidence and arguments presented
and do they address the main question posed?
Yes
6. Are the references appropriate?
Yes
Author Response
The manuscript described a model for dosing in the fasted state for simulation of the food effect. The manuscript, in general, is well written but some points should be considered:
- Please provide the ethics committee approval number and details about the guideline used for the clinical study.
Thank you for the comments. The details are given below:
Ethic Committee Name: National Research Ethics Service (NRES) Committee North East – York
Sponsor Reference Number: WP28978
EudraCT Number: 2014-001762-97
Approval date: 26-May-2014
The study was conducted in full conformance with the International Conference on Harmonisation (ICH) E6 guideline for Good Clinical Practice (GCP) and the principles of the Declaration of Helsinki. The study complied with the requirements of the ICH E2A guideline (Clinical Safety Data Management: Definitions and Standards for Expedited Reporting) and also with the European Union (EU) Clinical Trial Directive (2001/20/EC).
2. Please provide some details about the healthy adult volunteers.
The details have been added in the manuscript under the section 2.2. Clinical Pharmacokinetic Data of basmisanil
3. This study was built on a previous study published by Stillhart et al 2017. It is worth to conducted another study using another API with similar properties to validate the model.
Thank you for this comment. Yes we fully agree that more examples of this approach need to be shared and we are currently working on additional APIs.
Reviewer 2 Report
Dear authors
Nice presentation of your focused research
Can be extended a little more with additional studies, explanations, and the present use of the study
Thank you
Author Response
Thank you for this comment. Yes we fully agree that more examples of this approach need to be shared. We are currently working further on this model and on similar modeling for additional APIs. We hope to share more results next year.
Reviewer 3 Report
Modeling the effect of food on the pharmacodynamics of drugs is an important and in-demand area for pharmacology and pharmacy. The manuscript is of undoubted interest. At the same time, I suggest the authors to explain more clearly what caused the choice of basmisanil as the object of study.
Author Response
Thank you for this comment. We have added additional rationale for the food effect modelling of basmisanil in the introduction of the manuscript.